# Image Quality and Quantitative PET Parameters of Low-Dose [^18^F]FDG PET in a Long Axial Field-of-View PET/CT Scanner

**DOI:** 10.3390/diagnostics13203240

**Published:** 2023-10-18

**Authors:** Eduardo Calderón, Fabian P. Schmidt, Wenhong Lan, Salvador Castaneda-Vega, Andreas S. Brendlin, Nils F. Trautwein, Helmut Dittmann, Christian la Fougère, Lena Sophie Kiefer

**Affiliations:** 1Department of Nuclear Medicine and Clinical Molecular Imaging, University Hospital Tuebingen, 72076 Tuebingen, Germany; eduardo.calderon-ochoa@med.uni-tuebingen.de (E.C.); f.schmidt@med.uni-tuebingen.de (F.P.S.); wenhong.lan@med.uni-tuebingen.de (W.L.); salvador.castaneda@med.uni-tuebingen.de (S.C.-V.); nils.trautwein@med.uni-tuebingen.de (N.F.T.); helmut.dittmann@med.uni-tuebingen.de (H.D.); christian.lafougere@med.uni-tuebingen.de (C.l.F.); 2Werner Siemens Imaging Center, Department of Preclinical Imaging and Radiopharmacy, Eberhard-Karls University Tuebingen, 72076 Tuebingen, Germany; 3Department of Diagnostic and Interventional Radiology, University Hospital Tuebingen, 72076 Tuebingen, Germany; andreas.brendlin@med.uni-tuebingen.de; 4Cluster of Excellence iFIT (EXC 2180) “Image Guided and Functionally Instructed Tumor Therapies”, University of Tuebingen, 72074 Tuebingen, Germany; 5German Cancer Consortium (DKTK), Partner Site Tuebingen, 72074 Tuebingen, Germany

**Keywords:** total-body PET/CT scanner, LAFOV PET/CT, low-dose [^18^F]FDG PET, [^18^F]FDG

## Abstract

PET/CT scanners with a long axial field-of-view (LAFOV) provide increased sensitivity, enabling the adjustment of imaging parameters by reducing the injected activity or shortening the acquisition time. This study aimed to evaluate the limitations of reduced [^18^F]FDG activity doses on image quality, lesion detectability, and the quantification of lesion uptake in the Biograph Vision Quadra, as well as to assess the benefits of the recently introduced ultra-high sensitivity mode in a clinical setting. A number of 26 patients who underwent [^18^F]FDG-PET/CT (3.0 MBq/kg, 5 min scan time) were included in this analysis. The PET raw data was rebinned for shorter frame durations to simulate 5 min scans with lower activities in the high sensitivity (HS) and ultra-high sensitivity (UHS) modes. Image quality, noise, and lesion detectability (*n* = 82) were assessed using a 5-point Likert scale. The coefficient of variation (CoV), signal-to-noise ratio (SNR), tumor-to-background ratio (TBR), and standardized uptake values (SUV) including SUV_mean_, SUV_max_, and SUV_peak_ were evaluated. Subjective image ratings were generally superior in UHS compared to the HS mode. At 0.5 MBq/kg, lesion detectability decreased to 95% (HS) and to 98% (UHS). SNR was comparable at 1.0 MBq/kg in HS (5.7 ± 0.6) and 0.5 MBq/kg in UHS (5.5 ± 0.5). With lower doses, there were negligible reductions in SUV_mean_ and SUV_peak_, whereas SUV_max_ increased steadily. Reducing the [^18^F]FDG activity to 1.0 MBq/kg (HS/UHS) in a LAFOV PET/CT provides diagnostic image quality without statistically significant changes in the uptake parameters. The UHS mode improves image quality, noise, and lesion detectability compared to the HS mode.

## 1. Introduction

Positron emission tomography and computed tomography (PET/CT) hybrid imaging plays a fundamental role in current medical practice. Thereby, [^18^F]fluoro-deoxy-glucose (FDG) PET/CT has become an essential pillar in the diagnosis and follow-up of various oncological diseases, such as lymphoma, melanoma, and lung cancer [1,2,3]. Recent advances in the field of PET/CT technology have led to the introduction of PET/CT scanners with a long axial field-of-view (LAFOV), enabling the improvement of existing clinical applications and the establishment of new ones [4,5]. The extended axial field-of-view (aFOV) of these scanners, ranging between 64 to 200 cm, increases solid angle coverage for the detection of coincident events, providing a higher sensitivity and relevant improvements in the signal-to-noise ratio (SNR) compared to conventional PET scanners with a short axial field-of-view (SAFOV) [6,7]. Notably, the improved sensitivity is advantageous for examinations with low event statistics, such as PET imaging with radioisotopes that have low branching ratios such as Y-90 [8,9], allowing for post-therapeutic PET scans after trans-arterial radioembolization with Y-90 microspheres; immuno-PET imaging with long half-life radioisotopes such as Zr-89 [10,11], which allows the improved in vivo characterization of various tumors such as HER2-positive breast cancer [12]; and examinations with reduced doses of injected activity, contributing to a reduction in radiation exposure [13,14].

Conventional PET/CT scanners with a SAFOV of approximately 20 to 25 cm typically have a sensitivity in the range of 6 to 20 cps/kBq [15]. In these scanners, roughly 85 to 90% of the body is located outside the aFOV, and only around 3 to 5% of the available signal within the FOV can be used [16]. In contrast, the Biograph Vision Quadra PET/CT scanner (Siemens Healthineers, Knoxville, TN, USA) provides an aFOV of 106 cm and a sensitivity of 83 cps/kBq using its standard high-sensitivity (HS) mode with an acceptance angle of 18°. Compared to its predecessor model with a SAFOV (Biograph Vision 600), the Biograph Vision Quadra features an approximately fivefold sensitivity increase [17].

The ultra-high sensitivity (UHS) mode, representing the latest sensitivity mode for the Biograph Vision Quadra, allows for the unlimited acceptance angle of 52°, further improving the sensitivity to 176 cps/kBq [18]. This improvement translates into an approximately two-fold increase in sensitivity compared to the standard HS mode [19]. This enables a further reduction in acquisition time while maintaining the noise performance and might also provide advantages for the low-dose examination protocols while maintaining diagnostic reliability. Despite these encouraging results, the standardization of short acquisition time and low-dose protocols in clinical routine remains to be achieved. Specifically, to the best of our knowledge, no studies have yet been published systematically evaluating the limits of the reduction in injected activity regarding the qualitative and quantitative image parameters in the [^18^F]FDG PET/CT of oncological patients on the Biograph Vision Quadra system. Therefore, this study aimed to assess the impact of reduced [^18^F]FDG activity doses on image quality and noise, lesion detectability, and uptake quantification while comparing the HS and UHS modes in the Siemens Biograph Vision Quadra LAFOV PET/CT scanner.

## 2. Materials and Methods

### 2.1. Study Design and Patient Population

Twenty-six patients who underwent a whole-body [^18^F]FDG PET/CT on a Biograph Vision Quadra PET/CT scanner were included in this retrospective analysis. PET/CT examinations were clinically indicated for oncological purposes. Different tumor entities were included in this analysis (e.g., melanoma, lymphoma, lung cancer, esophageal cancer, colon cancer, rectal and anal cancer, and a cancer of unknown primary). Detailed patient characteristics are provided in Appendix A.

### 2.2. Imaging Protocol

Patients were required to fast for at least 10 h prior to the examination. A venous blood glucose measurement was conducted to confirm the blood glucose levels <130 mg/dL. A clinical standard dose of 3.0 MBq/kg [^18^F]FDG was injected intravenously, and PET/CT image acquisition was started at 60 min. p.i.. Whole-body scans were acquired in a supine position covering an area from the head to the mid-thighs. Technical details of the CT image acquisition and protocol are provided in Appendix A. The PET emission data with a routine acquisition time of 5 min were acquired directly after the CT scan.

### 2.3. Image Reconstruction

PET image reconstruction was performed using the proprietary e7 tools software (Version VR20, Siemens Healthineers, Knoxville, TN, USA). According to our standard clinical reconstruction protocol, an Ordinary-Poisson Ordered-Subsets Expectation-Maximization algorithm with four iterations and five subsets (OP-OSEM 4i5s) was applied, using point-spread-function (PSF) modeling and time-of-flight (TOF) information. The PET images were reconstructed with a matrix of 440 × 440 × 645 with a 1.65 × 1.65 × 1.65 mm^3^ isotropic voxel size, and no image filter was applied. Attenuation correction was performed based on the diagnostic CT scan acquired before the emission measurements. All reconstructions were performed for both the HS and the UHS modes. The PET raw data (5 min scan, 3.0 MBq/kg) was consecutively rebinned for shorter frame durations to simulate lower activities of injected [^18^F]FDG as follows: 1.0 MBq/kg, 0.5 MBq/kg, 0.25 MBq/kg, and 0.125 MBq/kg. An example of a data set reconstructed with the different simulated doses and both sensitivity modes is shown in Figure 1.

### 2.4. Image Analysis and Evaluation

Image analysis and evaluation were performed on a dedicated workstation using the SyngoVia^®^ software (Version 2.3.1, Siemens Healthineers; Knoxville, TN, USA) by two experienced resident nuclear medicine physicians (one being a board-certified radiology physician, both with multi-year experience in the reading of oncological PET/CT scans) in consensus. Both readers were blinded to the patient demographics and clinical information. Scans were read in multiple sessions in a randomized order. Afterward, all reconstructions per patient were re-read simultaneously to analyze the lesion uptake. In patients with a multifocal disease, a maximum of five lesions per patient were analyzed to reduce over-representation.

### 2.5. Subjective PET Image Quality

Subjective overall impression of image quality, subjective image noise, and conspicuity of suspected pathological lesions were assessed using a 5-point Likert scale. The criteria for the Likert scoring system are provided in Table 1.

### 2.6. Quantitative PET Analyses

Target lesion uptake was determined by placing a 40% isocontour volume of interest (VOI) around the lesion. Thereby, the mean, peak, and maximum standardized uptake values (SUV_mean_, SUV_peak_, and SUV_max_) were evaluated. As previously described, the liver was chosen to measure the background activity [20]. Background uptake was determined using a 14 cm^3^ VOI in the healthy tissue of the right liver lobe. Target lesions were manually segmented in the standard scan (3.0 MBq/kg, 5 min., HS mode). Afterwards, corresponding VOIs were automatically overlayed onto the simulated scans, ensuring that the placement of the respective VOIs was identical in all data sets.

The tumor-to-background ratio (TBR) was calculated as an estimate for the objective lesion contrast and lesion visibility as previously described [6], utilizing Equation (1):(1)$$TBR=\frac{SUVpeakLesion}{SUVmeanBackground}$$

Furthermore, to quantitatively assess image noise, the coefficient of variation (CoV) was calculated using Equation (2) as previously published [21]:(2)$$CoV=\frac{SUVSD}{SUVmean}$$

A CoV of <15% (as a recommended threshold by the EFOMP and EANM [22]) was considered for interpretation. According to [19], the SNR was defined as the reciprocal variable of (CoV) for the liver background. To account for physiological uptake differences between patients, the measured SUV from the different simulations was normalized to the corresponding SUV derived from the standard scan (3.0 MBq/kg, 5 min., HS mode). Subsequently, the normalized mean SUV values and standard deviations (SD) were determined.

### 2.7. Radiation Exposure

The effective radiation dose of the simulated examinations was calculated using the conversion factor of 0.019 mSv/MBq, as proposed by the International Commission on Radiological Protection publication 106 [23].

### 2.8. Statistical Analysis

Two-way ANOVA was performed to analyze the effect of the simulated reduced dose and sensitivity modes on the uptake values and SNR. Bonferroni’s multiple comparison tests were used to point out specific statistical significances following two-way ANOVA. The results of the two-way ANOVA are shown with *p*-values. An alpha level of 0.05 and a confidence interval of 95% were used for analysis. Corresponding degrees of freedom (df), F-values (F), and the results of multiple comparison tests are provided in Appendix A. *p* values < 0.05 were considered statistically significant. Statistical analysis was performed using GraphPad Prism (Version 9.4.1, GraphPad Software, San Diego, CA, USA).

## 3. Results

### 3.1. Overall PET Image Quality

The standard clinical scans with a dose of 3.0 MBq/kg in the HS mode were subjectively rated with a mean Likert score of 4.0 ± 0.4 (superior to the average quality). The reconstructed images with a dose reduction to 1.0, 0.5, 0.25, and 0.125 MBq/kg in the HS mode were subsequently rated with mean Likert scores of 3.1 ± 0.3, 2.0 ± 0.3, 1.0 ± 0.2, and 1.0 ± 0.0, respectively.

As shown in Figure 2A, the subjective image quality of image reconstructions in the UHS mode was generally rated superior compared to the images in HS mode. At full doses of 3.0 MBq/kg in the UHS mode, image quality was rated best, corresponding to “state-of-the-art quality” (Likert score 5.0 ± 0.0). The reconstructed images with the UHS mode received a Likert score of 4.1 ± 0.3 and 2.9 ± 0.3 for 1.0 and 0.5 MBq/kg, respectively, as well as 1.9 ± 0.3 and 1.1 ± 0.3 for 0.25 and 0.125 MBq/kg, respectively. However, the images at very low doses of 0.125 MBq/kg were not better rated as the images in the HS mode.

### 3.2. Detectability and Conspicuity of Suspected Pathological Lesions

In total, 82 lesions were selected in the standard reference scan with 3.0 MBq/kg in the HS mode and were considered for further analysis. Lesion detectability was not affected by the sensitivity modes in dose reductions down to 1.0 MBq/kg. The lesion detection rate decreased to 95%, 71%, and 49% (HS) and 98%, 83%, and 60% (UHS) for 0.5 MBq/kg, 0.25 MBq/kg, and 0.125 MBq/kg, respectively. The results of lesion detection rates for each simulated dose and sensitivity mode with the number of detected lesions are shown in Table 2.

Concerning subjective lesion conspicuity, all lesions were rated as “well-defined” at 1.0 MBq/kg with no notable different scoring between the HS and UHS mode (Likert score HS 3.8 ± 0.4 and UHS 4.1 ± 0.3), as shown in Figure 2B. At 0.5 MBq/kg in the HS mode, the lesions were “hazy but recognizable” (Likert score 3.0 ± 0.2). For 0.25 MBq/Kg in the HS mode, the lesions were considered “ill-defined, significantly impairing diagnostic confidence” (Likert score 2.0 ± 0.5). PET image reconstructions in the UHS mode at 0.5 MBq/Kg (Likert score 3.9 ± 0.3) and 0.25 MBq/kg (Likert score HS 3.2 ± 0.4) improved the lesion conspicuity. However, at the lowest simulated dose of 0.125 MBq/kg, the lesions were rated as mostly “unrecognizable” (Likert score HS 1.0 ± 0.0 and UHS 1.1 ± 0.3), independent of the sensitivity mode.

Regarding lesion morphology in the CT scan, the mean size of every undetected lesion was 1.5 ± 0.57 cm and comprised almost exclusively of small lymph nodes with a low to moderate [^18^F]FDG uptake (SUV_mean_ 4.7 ± 2.8). An example of a lesion being undetected in reconstructions with lower doses is provided in Figure 3.

### 3.3. Image Noise

Regarding subjective image noise ratings according to the Likert score (Figure 2C), the standard clinical images (3.0 MBq/kg, both in the HS and UHS mode) were considered to show a “near imperceptible noise”, corresponding to a Likert score of 5. Subjective image noise Likert scores of 4.8 ± 0.4, 4.0 ± 0.3, 3.0 ± 0.3, 2.1 ± 0.3, and 1.0 ± 0.2 (HS) and 5.0 ± 0.0, 4.3 ± 0.5, 4.0 ± 0.2, 3.2 ± 0.4, and 1.2 ± 0.4 (UHS) corresponded to 3.0, 1.0, 0.5, 0.25, and 0.125 MBq/kg, respectively.

At 1.0 MBq/kg in the HS mode, a mean CoV of 17.9 ± 2.0% was recorded, slightly surpassing the recommended CoV < 15.0% threshold. The mean CoV in the HS mode for 0.5 MBq/kg, 0.25 MBq/kg, and 0.125 MBq/Kg were 25.3 ± 2.6%, 35.5 ± 4.0%, and 49.0 ± 6.3%, respectively. For the UHS mode, the CoV was always lower; therefore, the same image noise could be obtained for UHS as for HS for lower doses, e.g., a CoV of 18.3% ± 1.8% (0.5 MBq/kg, UHS), which is comparable to 17.9 ± 2.0% with 1.0 MBq/kg in the HS mode. All mean values and standard deviations (SD) of the CoV and SNR measurements are provided in Table 3.

Two-way ANOVA on the SNR values revealed a statistically significant interaction between the effects of the sensitivity mode and simulated dose reduction (*p* < 0.0001). Furthermore, the results of the main effects showed that, in general, the sensitivity mode and simulated dose reduction independently have a statistically significant effect on the SNR values (*p* < 0.0001 for each main effect).

The Post-hoc multiple comparison test revealed that the mean values of SNR were significantly different between almost every simulated dose and sensitivity mode. This was, however, not the case between the reconstructions in the HS mode and the reconstructions in the UHS mode with half the simulated dose (e.g., 1.0 MBq/kg HS mode vs. 0.5 MBq/kg UHS mode, *p* > 0.99), as shown in Figure 4. Furthermore, a mean SNR UHS/HS ratio of all reconstructed images of 1.36 ± 0.02 was reported.

### 3.4. Quantitative PET Parameters

SUV_mean_ and SUV_peak_ only showed minimal decreases in the simulated low-dose reconstructions. Regarding the lowest dose of 0.125 MBq/kg, SUV_mean_ decreased by only 8%, whereas SUV_peak_ decreased by only 3%**.** The impact of the sensitivity mode on the SUV_mean_ and SUV_peak_ values was negligible. Two-way ANOVA revealed no statistically significant interaction between the effects of the sensitivity mode and the reduced doses, neither for SUV_mean_ nor SUV_peak_ (*p* > 0.99). The main effects analysis showed that neither sensitivity mode nor simulated dose reduction alone had a statistically significant effect on the SUV_mean_ or SUV_peak_ values (*p* = 0.76, *p* = 0.76, *p* = 0.93, and *p* = 0.81, respectively).

SUV_max_ increased steadily for lower doses. However, the increase was less prominent for UHS in comparison to the HS mode, e.g., we reported an increase of 16% (HS) and 6% (UHS) at 0.5 MBq/kg and 27% (HS) and 13% (UHS) at 0.25 MBq/kg.

Two-way ANOVA showed no statistically significant interaction between the effects of the sensitivity mode and the simulated dose reduction (*p* = 0.92) on SUV_max_. The main effects analysis shows that both the simulated dose reduction and the sensitivity mode had a statistically significant effect on the SUV_max_ values (*p* = 0.003, *p* = 0.04). Consistently, as shown in Figure 5C, post-hoc multiple comparison tests revealed that the significant difference in SUV_max_ values was present only when comparing the values between the standard scan (3.0 MBq/kg, HS mode) with 0.125 MBq/kg in the HS mode (*p* = 0.0135), where a mean increase of 41% was recorded. This was, however, not the case when comparing the standard scan SUV_max_ values (3.0 MBq/kg, HS mode) with values at 0.125 MBq/kg in the UHS mode (*p* > 0.99).

Regarding TBR (as a function of SUV_peak_), two-way ANOVA revealed no statistically significant interaction between the effects of the simulated dose reduction and the sensitivity mode (*p* > 0.99, *p* = 0.88). Moreover, the main effects analysis showed that neither the simulated dose reduction nor the sensitivity mode had a statistically significant effect on the TBR values.

## 4. Discussion

Recent studies have demonstrated the improved performance characteristics of LAFOV PET/CT scanners. For example, studies evaluating the uEXPLORER, another commercially available LAFOV PET/CT scanner with an aFOV of approximately 2 m, showed that low- (1.85 MBq/kg) or ultra-low-dose (0.37 MBq/kg) imaging protocols [24,25] may be clinically feasible, usually requiring longer scan times (>5 min.).

For the Biograph Vision Quadra, a study in melanoma patients showed that a dose reduction down to 2.0 MBq/Kg with a 5 min scan was not associated with a worsening clinical performance [14]. Such protocols may profit from the recently introduced UHS mode, which has the potential to further lower the applicated activity and/or the acquisition time by fully exploiting the largely increased sensitivity of this LAFOV PET/CT scanner. However, before the low-dose protocols involving the latest sensitivity mode can be implemented into routine clinical practice, evaluating their limitations and diagnostical reliability is necessary. Therefore, we assessed the influence of reduced [^18^F]FDG activity doses for both the HS and UHS modes on the qualitative and quantitative PET parameters in the Biograph Vision Quadra.

Regarding subjective image rating, we determined that the images at 0.5 MBq/kg (1/6th of the standard clinical dose) with a 5 min acquisition in the HS mode were “barely diagnostic”. In this regard, simulations with very low doses in the HS mode (0.25 MBq/kg and 0.125 MBq/kg) were rated as biased via the poor image quality and affected diagnostic reliability. However, using the same protocols in the UHS mode, improved diagnostic image quality was consistently achieved, e.g., the images at 0.25 MBq/kg in the UHS mode were then also rated as “barely diagnostic”. Nonetheless, we would still recommend that these very low doses should not be considered for clinical application unless compensated by a scan time considerably longer than 5 min. In general, the enhanced SNR in the UHS mode led to an overall improvement in the subjectively evaluated image characteristics, except for 0.125 MBq/kg, where no tangible improvements were noted.

Lesion detectability decreased at the dose of 0.5 MBq/kg to a detection rate of 95% in the HS mode and could be partly recovered to 98% in the UHS mode. Lesion morphology and tracer uptake evidently influenced lesion detection. The mean size of undetected lesions was 1.5 ± 0.57 cm with a SUV_mean_ of 4.7 ± 2.8, mainly indicating an increased risk of smaller and low to moderate metabolic active lesions remaining undetected.

A dose of 1.0 MBq/kg combined with the HS mode resulted in a CoV of 17.9% ± 2.0, exceeding the threshold of 15.0% recommended by EFOMP and EANM [22]. The UHS mode could meet this criterion with a CoV of 13.0 ± 1.3%. To further reduce the dose and maintain an acceptable image noise, filtering of the images can be performed. In this regard, Rausch et al. [26] showed that for the Biograph Vision Quadra, using a 6 mm Gaussian filter instead of a 2 mm filter helped to increase the useful aFOV (CoV < 15%) from 83 cm to 103 cm. Subjective image noise ratings considered noise as excessive and impairing the diagnostic quality starting at 0.25 MBq/Kg while having acceptable image noise levels even at 0.5 MBq/kg, implying that higher noise values may still be acceptable in clinical settings.

Furthermore, two-way ANOVA showed that the SNR values of the images reconstructed in the UHS mode have comparable SNRs to the images reconstructed in the HS mode but with twice the simulated activity, e.g., 0.5 MBq/kg (UHS) vs. 1.0 MBq/kg (HS), which is another way of interpreting the similar results previously published, where 30 s/60 s acquisitions in UHS showed comparable SNRs to 60 s/120 s acquisitions in the HS mode [19]. Furthermore, a mean SNR UHS to the HS ratio of 1.36 ± 0.02 was reported, which also aligns with the previously published values. The UHS mode is, thus, useful for acquiring clinically diagnostic images with lower activity protocols.

Of note, image noise is not uniformly improved across the FOV. Schmidt et al. [27] reported that for the Biograph Vision Quadra, the improvement in CoV in the center FOV can be as high as a factor of 1.49, comparing the UHS and HS mode. However, no improvement was observed with the UHS mode beyond the central 80 cm of the aFOV.

Concerning the quantitative PET parameters, SUV_mean_, SUV_peak_, and TBR showed only minimal decreases of <10% and two-way ANOVA revealed no statistically significant differences, including at the lowest simulated dose. SUV_max,_ however, was steadily considerably higher in the HS mode compared to the values in the UHS mode and increased towards lower doses, which is comparable to the previously published results, where 600 s acquisitions had statistically significant differences in the SUV_max_ values of evaluated lesions compared to 120 s and 60 s acquisitions [28], whereas SUV_mean_ and SUV_peak_ remained relative stable. This is expected to be caused by the influence of noise on SUV_max_ [29,30]. In our study, two-way ANOVA and multiple comparisons revealed only statistically significant differences while comparing the SUV_max_ values of the standard 3.0 MBq/kg scan (HS) with values at 0.125 MBq/kg (HS), whereas when comparing to SUV_max_ values at 0.125 MBq/kg in UHS mode two-way ANOVA showed no statistically significantly differences. SUV_max_ is known to be overestimated due to the PSF reconstruction [31,32] and prone to noise distortion as it is based on the value of only a single voxel. Therefore, using SUV_peak_ or SUV_mean_ for quantification instead of SUV_max_ seems reasonable, especially in scenarios with low event statistics. The fact that the uptake values remained relatively constant at lower doses down to 0.125 MBq/kg (=1/24th of the standard applied dose, UHS mode) indicates that the fundamental limitations regarding dose reduction were based on insufficient diagnostic image quality.

Considering the results of this study, the Biograph Vision Quadra and LAFOV PET/CT scanners, in general, provide an emerging flexibility in the adjustment of applied activities and scan times. Of note, instead of altering the dose, a change in scan duration can also have a comparable impact. As such, considering the short half-life of F-18 of 109.8 min, a 5 min scan with 1.0 MBq/kg would yield approximately the same count statistics as a 10 min scan with 0.5 MBq/kg. As with 1.0 MBq/kg, no degradations were reported in our work compared to the full 3.0 MBq/kg with a 5 min scan. This implies the same is true for 0.5 MBq/kg with a 10 min scan time. This would allow high image quality, quantification accuracy, and diagnostic reliability with an effective dose of less than 1 mSv (for a 70 kg patient).

Utilizing this protocol with 0.5 MBq/kg and a 10 min scan would also allow feasible dual-tracer same-day protocols; for example, by performing an [^18^F]FDG PET early in the day. Approximately three half-lives later, one could perform a second PET scan with a different radiotracer and standard activity doses (2.0–3.0 MBq/kg) of, e.g., [^18^F]PSMA-1007 or [^18^F]SiFAlin-TATE, since no relevant activity of [^18^F]FDG remains at the time point of the second scan. Such protocols would come into question where the [^18^F]FDG PET supports clinical decision-making; for example, for patient selection or therapy control in patients with prostate cancer or neuroendocrine tumors that may receive or are receiving radioligand therapy. This type of examination aims to find radiotracer mismatches (FDG-positive and PSMA/SSR negative), which may indicate more aggressive tumor types that may warrant more aggressive therapy regimens instead of radioligand therapy [33]. The benefit of this protocols compared to previously published ones, e.g., first completion of [^68^Ga]Ga-PSMA-11 PET scan followed immediately by a standard 40 MBq injection of [^18^F]FDG [34], is that the semi-quantitative assessment of both FDG and PSMA would be less problematic at the cost of longer patient waiting times. Nonetheless, both protocols would be able to assess radiotracer mismatches.

Considering the reduced radiation exposure, pediatric patients could also benefit, and hybrid imaging could become more feasible in this patient population. The flexibility of LAFOV PET/CT scanners comes to note in pediatric patients since, depending on the clinical setting, one would be able to apply a standard activity dose and perform a fast scan while avoiding the use of anesthesia, as Reichkendler M et al. showed [35], or reduce applied activity in children that do not require sedation. Our results might be transferable for hybrid imaging studies in the pediatric population, but this requires further evaluation.

One limitation of this study is that only a relatively small sample size of 26 patients with a total of 82 pathological target lesions was included for analysis. Only two patients included were obese with a BMI > 35 kg/m^2^. Previous studies reported that a dose reduction was feasible only in non-obese patients with a BMI < 30 kg/m^2^, thus limiting the generalizability of our results to the obese patient population [36]. Furthermore, different FDG-avid tumor entities were included in this study, while other entities were not represented in our clinical cohort. We recognize that this might limit the generalizability of our results. Further studies including low-metabolic tumors are needed to further evaluate the performance of low-dose [^18^F]FDG PET in these entities. However, we believe that our study provides a satisfactory foundation for possible future evaluations in this regard. Another limitation is that only [^18^F]FDG in oncological patients was considered in this study, and no non-oncological indications for PET/CT (e.g., search for florid inflammation or fever of unknown origin) were analyzed. Moreover, the transferability to tracers based on isotopes different than F-18, which have a larger positron range (such as Ga-68) or typically require lower activities (such as Zr-89), need to be elaborated as these factors impact the quantification accuracy and noise performance [37]. Further studies are warranted to assess the feasibility of low-dose PET examinations in these scenarios.

## 5. Conclusions

Due to their high sensitivity, LAFOV PET/CT scanners allow for a considerable reduction in the applied activity of radiotracers. In this study, we showed that reducing the injected [^18^F]-FDG activity to 1.0 MBq/kg in a LAFOV PET/CT still offered clinically diagnostic image quality without relevant changes in the uptake parameters. Further reductions appear feasible by prolonging the scan times (e.g., 10 min) and by using the UHS mode. Fundamental limitations of further dose reduction were based on insufficient diagnostic image quality, the diminishing performance, and did not involve any possible alterations in the quantitative uptake parameters. These novel insights may further expand the implementation in clinical routine, possibly allowing for e.g., same-day multi-tracer protocols and applications in pediatric patient populations.

## Figures and Tables

**Figure 1 diagnostics-13-03240-f001:**
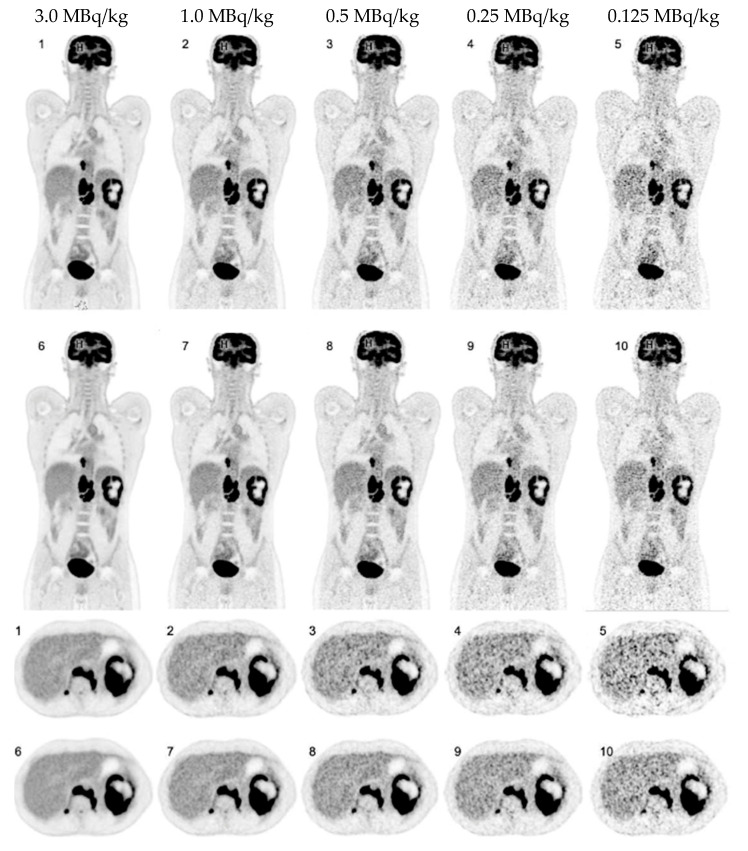
Coronal and axial [^18^F]FDG PET images of a 37-year-old patient with Hodgkin lymphoma reconstructed in high-sensitivity (1–5) and ultra-high sensitivity (6–10) mode.

**Figure 2 diagnostics-13-03240-f002:**
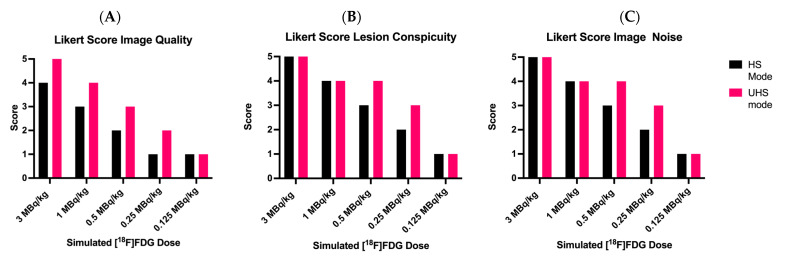
Results of subjective PET image quality ratings. (**A**): Overall Image Quality; (**B**): Lesion Conspicuity; and (**C**): Image Noise. UHS: ultra-high sensitivity; HS: high sensitivity.

**Figure 3 diagnostics-13-03240-f003:**
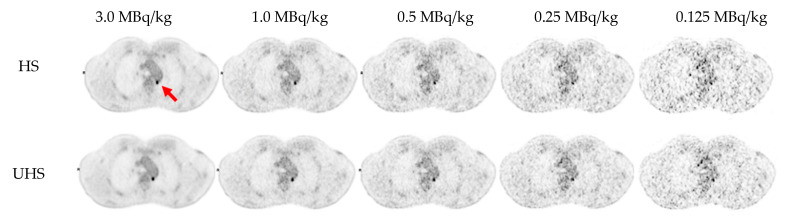
Example of a small retro-aortic lymph node (red arrow, 7 × 6 mm^2^) undetected in reconstructions with lower doses (0.25 MBq/kg in HS mode and 0.125 MBq/kg in both HS and UHS mode). UHS: ultra-high sensitivity; HS: high sensitivity.

**Figure 4 diagnostics-13-03240-f004:**
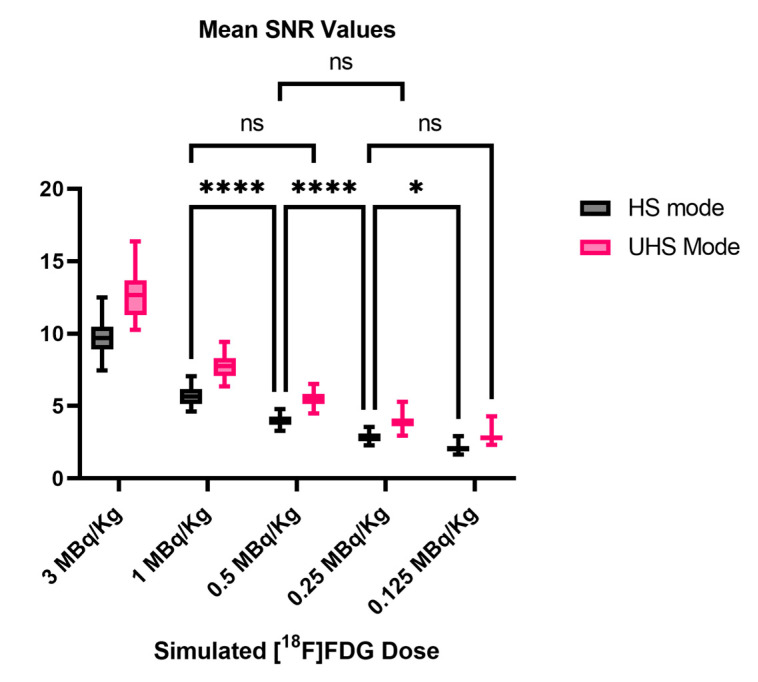
Mean SNR values of liver background according to simulated dose and sensitivity mode. SNR values were statistically significantly different between almost all reconstructed images. Dose reduction and sensitivity mode had relevant effects on SNR values. However, in UHS images, SNR was not statistically significantly different to values in HS images but with twice the dose (e.g., 0.5 MBq/kg UHS vs. 1.0 MBq/kg HS). Each boxplot shows the median value (central line) and the 25–75th percentiles. Whiskers represent the minimum and maximum values. Two-way ANOVA was performed for statistical analysis followed by Bonferroni’s multiple comparison test. Not every comparison between every simulated dose and sensitivity mode is displayed to avoid an overcrowded graph. * Refers to *p* < 0.05 and **** refers to *p* < 0.0001. HS: high sensitivity; UHS: ultra-high sensitivity; SNR: signal-to-noise ratio; ns: non-significant.

**Figure 5 diagnostics-13-03240-f005:**
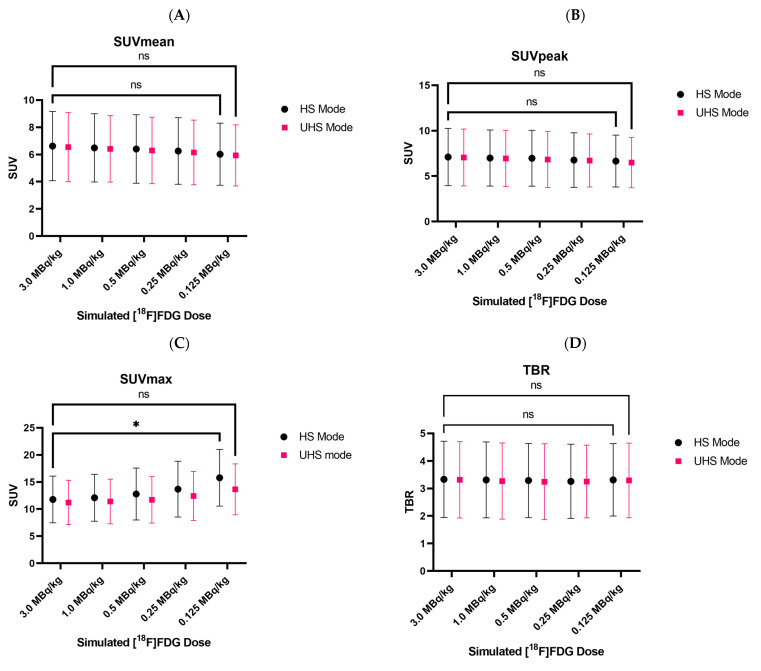
Average standard uptake values of all lesions for SUV_mean_ (**A**), SUV_peak_ (**B**), SUV_max_ (**C**), and TBR (**D**) for all simulated doses in both sensitivity modes. No statistically significant differences in SUV_mean_, SUV_peak_, and TBR values were reported between the standard scan (3 MBq/kg, HS mode) and all simulated doses, regardless of sensitivity mode and even at the lowest dose. SUV_max_ values were, however, statistically significantly different between the standard scan and simulated dose of 0.125 Mbq/kg in HS mode (**C**). Each dot represents the mean value and error bars represent the standard deviations. Two-way ANOVA was performed for statistical analysis, followed by Bonferroni’s multiple comparison test. Not every comparison between every simulated dose and sensitivity mode is displayed to avoid an overcrowded graph. * Refers to *p* < 0.05. UHS: ultra-high sensitivity: HS, high sensitivity; STD: standard deviation; AU: arbitrary unit; ns: non-significant.

**Table 1 diagnostics-13-03240-t001:** 5-Point Likert scoring system for subjective PET image rating.

Score	Image Quality	Lesion Conspicuity	Image Noise
5	state-of-the-art quality	well-defined	near-imperceptible noise
4	superior to the average	fairly defined	lower than regular image of daily practice
3	regular quality of daily practice	hazy, recognizable	similar to regular image of daily practice
2	barely diagnostic	ill-defined, impairing diagnostic confidence	increased noise, slightly worse than regular image of daily practice
1	non-diagnostic	un-recognizable	excessive noise

**Table 2 diagnostics-13-03240-t002:** Lesion detection rate according to simulated dose and sensitivity mode.

[^18^F]FDG MBq/kg	Sensitivity Mode	Number of Lesions Detected	Lesion Detection Rate in %
3.0	UHS	82	100%
3.0	HS	82	100%
1.0	UHS	82	100%
1.0	HS	82	100%
0.5	UHS	80	98%
0.5	HS	78	95%
0.25	UHS	68	83%
0.25	HS	58	71%
0.125	UHS	49	60%
0.125	HS	40	49%

UHS: ultra-high sensitivity; HS: high sensitivity.

**Table 3 diagnostics-13-03240-t003:** Coefficient of Variation and signal-to-noise ratio mean values according to simulated dose and sensitivity mode.

[^18^F]FDG MBq/kg	Sensitivity Mode	Mean CoV and STD in %	Mean SNR and STD
3.0	UHS	7.9 ± 1.0	12.8 ± 1.6
3.0	HS	10.4 ± 1.2	9.7 ± 1.1
1.0	UHS	13.0 ± 1.3	7.8 ± 0.8
1.0	HS	17.9 ± 2.0	5.7 ± 0.6
0.5	UHS	18.3 ± 1.8	5.5 ± 0.5
0.5	HS	25.3 ± 2.6	4.0 ± 0.4
0.25	UHS	26.0 ± 3.3	3.9 ± 0.5
0.25	HS	35.5 ± 4.0	2.9 ± 0.3
0.125	UHS	35.7 ± 4.5	2.9 ± 0.4
0.125	HS	49.0 ± 6.3	2.1 ± 0.3

UHS: ultra-high sensitivity; HS: high sensitivity; CoV: coefficient of variation; STD: standard deviation; SNR: signal-to-noise ratio.

## Data Availability

The data presented in this study are available on request from the corresponding author. The data are not publicly available due to privacy and ethical reasons.

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
