# Peer review of "Image Quality and Quantitative PET Parameters of Low-Dose [18F]FDG PET in a Long Axial Field-of-View PET/CT Scanner"

_diagnostics, 2023, doi:10.3390/diagnostics13203240_

Round 1
Reviewer 1 Report
Thank you for providing data concerning such a novel topic, of very high interest nowadays.
I have some minor comments.
- It would be much preferable to substitute the words "total-body PET/CT" with "Long axial field of view (LAFOV) PET/CT" in the whole paper, as it would be the preferable definition for the Quadra scanner and any other scanner different from the 194 cm commercially available scanner.
- In Figures 1, 2 and 3, the headings of each column should stay near the figure. This might not depend on the Authors, but if they can make sure that the headings follow the figures independently from the paper formatting, this would help the readability of the figures themself.
Author Response
We thank you for the critical evaluation of our work and for the helpful suggestions and questions as well for the opportunity to revise our manuscript. We have addressed the raised points as detailed below.
Reviewer #1
Comment #1: It would be much preferable to substitute the words "total-body PET/CT" with "Long axial field of view (LAFOV) PET/CT" in the whole paper, as it would be the preferable definition for the Quadra scanner and any other scanner different from the 194 cm commercially available scanner.
Thank you for pointing this out. We agree with the reviewer and have replaced the acronym and definition accordingly in the whole manuscript.
Comment #2: In Figures 1, 2, and 3, the headings of each column should stay near the figure. This might not depend on the Authors, but if they can make sure that the headings follow the figures independently from the paper formatting, this would help the readability of the figures themselves.
We thank the reviewer for this important comment and have changed the format as possible.
Reviewer 2 Report
1- Define all abbreviations at their first presence in the text.
2- All equations should be numbered and referred to the corresponding number in the text.
3- The manuscript has no appropriate description of the results. All tables and figures should be described in the results section and data should be compared statistically.
4- If it is possible, please provide demographic information of patients in the text.
5- Figure legends should be provided self-explanatory in detail. Also, all abbreviations used in tables should be defined in figure legends.
6- Minor editing of English language grammar and spelling is required.
Minor editing of English language grammar and spelling is required.
Author Response
We thank you for the critical evaluation of our work and for the helpful suggestions and questions as well for the opportunity to revise our manuscript. We have addressed the raised points as detailed below.
Reviewer #2
Comments #1-3: 1. Define all abbreviations at their first presence in the text. 2. All equations should be numbered and referred to the corresponding number in the text. 3. Figure legends should be provided self-explanatory in detail. Also, all abbreviations used in tables should be defined in figure legends.
Thank you very much for bringing our attention to these important details. Changes have been made accordingly throughout the whole manuscript.
Comment #4: The manuscript has no appropriate description of the results. All tables and figures should be described in the results section, and data should be compared statistically.
We thank the reviewer for this critical comment. The main text has been revised to better describe the results while also adding new figures to better denote statistical significance and the contrary.
Comment #5: If it is possible, please provide demographic information of patients in the text.
Thank you for raising this point. Demographic information regarding the different tumor entities has been added accordingly in the methods section.
Comment #6: Minor editing of English language grammar and spelling is required.
We thank the reviewer for this comment and have revised the manuscript accordingly.
Reviewer 3 Report
The authors described 26 patients 20 who underwent [18F]FDG-PET/CT (3.0 MBq/kg) scans. The results were simulated with lower activities in high sensitivity (HS) and ultra-high sensitivity (UHS) modes. They found the detection rate and quality were comparable at 1.0 MBq/kg in HS and 0.5 MBq/kg in UHS, in comparison to 3.0 MBq/kg.
Their findings are quite novel and may bring about change in practice.
I have several comments/concerns.
1. The author should provide the reason for using 3.0 MBq/kg in the first place. It appears at 1.0 MBq/kg or even 0.5 MBq/kg is sufficient.
2. With FDG 0.25 MBq/kg, the detection rate (sesnsitivity) became lower. It could be related to cancer type. The authors should report the reduced sensitivity of specific cancer types.
3. In fact, every cancer type may be different. This report put several cancers (lymphoma, melanoma, squamous cell ca) together. In a preliminary report it is perhaps fine. However, they should describe some difference in cancer types and they may suggest expanding the study scale for specific cancer types in order to make new standards.
4. The authors suggest further reduction is possible by prolonging scan time (10 min). Will that make patients exposed to higher radiation dose?
Author Response
We thank you for the critical evaluation of our work and for the helpful suggestions and questions as well for the opportunity to revise our manuscript. We have addressed the raised points as detailed below.
Reviewer #3
The authors described 26 patients 20 who underwent [18F]FDG-PET/CT (3.0 MBq/kg) scans. The results were simulated with lower activities in high sensitivity (HS) and ultra-high sensitivity (UHS) modes. They found the detection rate and quality were comparable at 1.0 MBq/kg in HS and 0.5 MBq/kg in UHS, in comparison to 3.0 MBq/kg. Their findings are quite novel and may bring about change in practice. I have several comments/concerns.
Comment #1: The author should provide the reason for using 3.0 MBq/kg in the first place. It appears at 1.0 MBq/kg or even 0.5 MBq/kg is sufficient.
We thank the reviewer for this critical comment. In fact, this retrospective study was based on clinically indicated routine scans that were performed according to the current standards at our PET/CT site in a clinical setting. At that time, scans were acquired with a standard clinical dose of 3 MBq/kg [18F]FDG. In this context, we aimed to evaluate image quality and performance of dose reduction in comparison to our standard clinical images at the above-mentioned dose.
Comment #2 & #3: 2. With FDG 0.25 MBq/kg, the detection rate (sensitivity) became lower. It could be related to cancer type. The authors should report the reduced sensitivity of specific cancer types. 3. In fact, every cancer type may be different. This report put several cancers (lymphoma, melanoma, squamous cell ca) together. In a preliminary report, it is perhaps fine. However, they should describe some differences in cancer types, and they may suggest expanding the study scale for specific cancer types in order to make new standards.
Thank you very much for this very important comment. In fact, our aim was to evaluate a broad selection of FDG-avid tumor entities to be able to generalize the results as much as possible. We agree with the reviewer, that the results cannot be generalized to all tumor entities. Therefore, we have included a pertaining phrase in the limitations section of our manuscript, also highlighting the need for further, entity-specific studies. As stated in the results part, the mean size of undetected lesions was small (1.5 ± 0.2 cm) with a variable, low to moderate, FDG uptake. This might imply that the risk of under-staging lies primarily with low metabolic entities and small metastasis/lesions
Comment #4: The authors suggest further reduction is possible by prolonging scan time (10 min). Will that make patients exposed to higher radiation doses?
Thank you very much for this important question. In fact, prolonging scan time would not mean a higher radiation dose for the patient since this only implies a longer emission scan time is being performed while having a fixed amount of [18F]FDG being intravenously applied 60 minutes before (e.g. 0.5 MBq/Kg of [18F]FDG, with a 10 min. emission scan time corresponding to 1 MBq/Kg and 5 min. emission scan).